# Psychosocial Mechanism of Adolescents’ Depression: A Dose-Response Relation with Physical Activity

**DOI:** 10.3390/children7040037

**Published:** 2020-04-24

**Authors:** Man Xiang, Xiangli Gu, Xiaoxia Zhang, Samantha Moss, Chaoqun Huang, Larry Paul Nelson, Tao Zhang

**Affiliations:** 1College of Public Health, Zhejiang Institute of Mechanical and Electrical Engineering, Hangzhou 310053, China; xiangman@zime.edu.cn; 2Department of Kinesiology, the University of Texas at Arlington, Arlington, TX 76019, USA; xiaoxia.zhang@mavs.uta.edu (X.Z.); samantha.moss2@mavs.uta.edu (S.M.); lnelson@uta.edu (L.P.N.); 3Department of Exercise and Sport Science, Wayland Baptist University, Plainview, TX 79072, USA; huangc@wbu.edu; 4Department of Kinesiology, Health Promotion and Recreation, University of North Texas, Denton, TX 76203, USA; tao.zhang@unt.edu

**Keywords:** dose-response, self-efficacy, depression, physical activity, middle schools, youth

## Abstract

Depression has become the most prevalent mental health problem in developing countries, and especially among adolescents. Lubans and his colleagues proposed a psychosocial mechanism to understand the trajectory of mental health (i.e., depression). Thus, this study aimed (1) to examine the relations between different doses of physical activity (PA), light PA (LPA), moderate PA (MPA), and vigorous PA (VPA), academic self-efficacy, and depression among adolescents, and (2) to investigate the direct and indirect relations of various doses of PA to depression through academic self-efficacy among middle school adolescents. Participants were 428 (235 boys, Mean *_age_* = 13.7) adolescents recruited from two middle schools in China. They completed previously validated questionnaires to measure different intensity levels of PA (LPA, MPA, and VPA), academic self-efficacy, and depression. There were significant associations of academic self-efficacy with three different doses of PA (*p* < 0.01). Both LPA and MPA were negatively associated with depression but not VPA. Structural equation modeling (SEM) revealed a well-fit model suggesting the psychosocial pathway from different doses of PA to depression through academic self-efficacy. Findings of this study indicated that academic self-efficacy regulates adolescents’ depression. Tailoring different intensities of PA benefits adolescents’ academic self-efficacy by framing the positive and supportive environment in schools, which can potentially reduce the prevalence of depression during adolescence.

## 1. Introduction

Depression is among the most common mental disorders worldwide [1,2], while depression in adolescents is common but often unrecognized. People with depression experience various symptoms including irritability, loss of interest, decreased energy, feelings of sadness and worthlessness, thoughts of death or suicide [3]. The symptoms must be present most of the day, nearly every day, for at least two weeks [4]. The recent statistical report showed that one in five adolescents in the United States [5] and over 24.3% of youth in China are suspected of depressive symptoms [6]. As a major risk factor for suicide among adolescents [7], depression leads to severe social and educational impairments [8,9] and increases the rate of smoking, substance misuse, and obesity [10,11]. Unfortunately, more than 60% of American adolescents with depressive symptoms did not receive treatment (e.g., medication or health care services) [4]. Given the fact that depression in adolescence is likely to trigger other mental health problems in adulthood [12], it has been a priority for public health to examine effective coping strategies regarding depression prevention and treatment.

In 2016, Lubans and his colleagues proposed a conceptual model to hypothesize the psychosocial mechanism to understand the relationship between physical activity (PA) and mental health outcomes (i.e., cognition, well-being, and ill-being) among children and adolescents. The psychosocial mechanism proposes that PA participation would impact individuals’ psychosocial perceptions and, ultimately, influence their mental health outcomes [13]. In other words, the mental health benefits of participating in PA may get additional benefit through social interactions (i.e., increased self-efficacy, and perceived competence) among youth. In Lubans and colleagues’ review [13], 14 out of 18 empirical studies found that different PA/exercise dominated intervention/experiments resulted in a positive influence on mental health outcomes (primarily self-esteem = 11, depression = 4, quality of life = 3), and proposed that this relationship may be mediated by various psychosocial indicators such as perceived physical self-concept, perceived competence, and perceived physical appearance [13]. Although most of the reviewed studies only focused on the direct association between PA and mental health outcomes, the potential mediators or the psychosocial mechanism of this association have not been investigated among youth [13].

It is well-documented that PA participation has a positive association with depression in both adolescents [13,14,15] and adults [16,17]. The randomized controlled trials (RCT) among adults revealed that at least moderate-to-vigorous intensity levels of PA (MVPA) have significant positive effects in people with depression [16]. A recent review based on the RCTs (11 studies) noted that a light-to-moderate PA but not vigorous level of PA (VPA), three times per week for 6–12 weeks, would improve depressive symptoms in a clinical sample of adolescents [14]. Carter and colleagues also suggest that a preferred level of PA (i.e., low-to-moderate intensity) can lead to improved mood and enjoyment, which, ultimately, results in the reduction of depressive symptoms among adolescents who were in treatment for depression (*n* = 26) [18]. Similar results in another study found preferred intensity of PA (i.e., LPA) leading to the reduction of depression [19]. However, the study in pre-adolescents revealed that individuals engaged in more MVPA at both age 6 and age 8 years demonstrated fewer depressive symptoms two years later compared to their counterparts [20]. In addition, depressed adolescents (12–18 years) who engaged in VPA demonstrated a rapid reduction of depression than those involved in LPA in 6–8 weeks of intervention, but no significant differences were observed after a 12-week intervention [21].

It remains unclear to what extent the dose responses of PA with depression among healthy adolescents has and how this relationship may be modified through psychosocial indicators, such as perceived competence and/or self-efficacy (i.e., underlying psychosocial mechanisms). Researchers found that the relations between PA and self-efficacy were different among children with and without depression [22]. The reverse associations between self-efficacy and depression were also documented among adolescents, which suggests that higher self-efficacy is associated with less depressive symptoms [23,24]. For instance, Annesi found that, by offering pre-adolescents after-school PA programs, the increases of children’s self-concept were associated with decreases of their depressive symptoms [25]. Another empirical research study revealed that the psychosocial factor (i.e., global self-worth) emerged as a mediator in the effects of PA toward depression among 7–11 years old children who are overweight [26]. The findings of the study found that engaging in more MVPA (>20 min) had more deduction in depressive symptoms and higher enhancement of self-worth compared to control groups [26]. Furthermore, one recent study provided preliminary evidence regarding the PA dose responses toward self-efficacy and found no direct relationship between VPA and self-efficacy among adolescents [27]. To date, the current literature of depression has provided minimal evidence about the optimal intensity (dose) level of PA toward depression reduction [14] and how this relationship would deviate through the mediating role of self-efficacy [26].

Guided by the conceptual model proposed by Lubans and colleagues [13], the major purpose of the current study was to test PA dose-response toward depression and to investigate whether different intensity levels of PA (LPA, MPA, and VPA) and self-efficacy synergize, interfere, or compensate for one another to predict depression in adolescents [13]. Specifically, this study aimed to (1) examine the relations among different doses of PA (LPA, MPA, and VPA), academic self-efficacy, and depression in adolescents, and (2) investigate the direct and indirect relations of various doses of PA with depression through academic self-efficacy (test the psychosocial mechanism). It is hypothesized that there would be significant associations between LPA, MPA, VPA, academic self-efficacy, and depression among adolescents. Different doses of PA would have different direct and indirect predictive strengths toward depression through academic self-efficacy among adolescents.

## 2. Materials and Methods

This study protocol was approved by the university institutional review board (IRB No. 16-558), and the support and approval was also granted by the school principals and physical education (PE) coordinators. Parental informed consent forms and child assent forms were obtained in accordance with the participating school district and the Declaration of Helsinki before data collection.

### 2.1. Participants

Data for this study was collected from two middle schools in the east region of China. A total of 428 adolescents (235 boys, 193 girls, Mean *_age_* = 13.7 ± 1.5) were recruited in which 49.1% of them were in 7th grade and the rest of them were in 8th grade.

### 2.2. Procedures

The original instruments were in English. After being translated into a Chinese version, a back-translation to English was also applied to ensure the accuracy of the instrument. The data collection was initialized during PE classes after the parental informed consent forms and child assents were received. Administrated by research assistants, participants were asked to fill a previous validated survey assessing their PA, academic self-efficacy, depressive symptoms, and basic demographic information such as age and gender. The questionnaires were collected immediately after completion in classes.

### 2.3. Measures

#### 2.3.1. Physical Activity 

Participants’ PA was assessed using the Leisure Time Physical Activity Scale (LTPA) [28]. Participant self-reported four items about the frequency in a typical week (ranges from 0–8) that one engages in LPA, MPA, and VPA during leisure time for at least 15 min. Based on the scoring guideline [28], the three intensity levels of PA were based on the corresponding Metabolic Equivalent of Task (MET) value (LPA = 3 MET, MPA = 5 MET, and VPA = 9 MET). Specifically, each intensity level of PA was calculated as: LPA = frequency × 3 MET, MPA = frequency × 5 MET, and VPA = frequency × 9 MET, respectively. The calculated scores of LPA (ranges 0–24), MPA (ranges 0–40), and VPA (ranges 0–72) were used in the data analysis. This questionnaire has been widely used in assessing Chinese adolescents’ PA with good reliability and validity [29].

#### 2.3.2. Academic Self-Efficacy

The 5-item academic self-efficacy was adapted to assess participants’ academic self-efficacy [30]. Participants were asked to rate five items using a seven-point Likert-type scale ranging from 1 (very disagree) to 7 (very agree). The items include: (1) I am confident in my scholastic ability, (2) I do well in school, (3) I learn new concepts quickly, (4) I am successful, and (5) I am confident in my ability to succeed in school. The final score of academic self-efficacy is the average score from the five items. Thus, higher values indicate higher levels of academic self-efficacy. The Cronbach’s alpha coefficient for this scale was 0.94 in the current study.

#### 2.3.3. Depressive Symptoms

The Chinese version of the 20-item Center for Epidemiological Studies-Depression Scale (CES-D) [31] was applied to assess the depressive symptoms among participants. Participants respond to items regarding how often they have felt or behaved during the past week, such as “I thought my life has been a failure.” The four-point Likert scale that was used for each statement ranges from 0 (not at all) to 3 (a lot). The depression score is the summary of the 20-item (0–60) in which the higher score indicates a higher risk of depression. The CES-D scale has been validated in previous studies [32] and demonstrated good reliability for children and adolescents [33,34]. The Cronbach’s alpha coefficient of the CES-D was 0.87 in the current study.

### 2.4. Statistical Analysis

Because the missing data was random and less than 5%, a series mean replacement was employed before data analysis. The SPSS 25.0 version was used to conduct the descriptive statistics, internal consistency, and bivariate correlations among study variables. Structural Equation Modelling (SEM) was conducted using AMOS 25.0. Before structuring the model, a confirmatory factor analysis (CFA) was conducted to examine the construct validity and internal reliability in the current study. After that, a hypothesized SEM model was structured to test the direct and indirect effects of different doses of PA (LPA, MPA, and VPA, exogenous variables) on depression (endogenous variable) through academic self-efficacy (latent/exogenous variable). The maximum-likelihood estimation was applied for the analyses. Multiple indices were applied to evaluate the goodness of the model fit: Chi-Square goodness-of-fit test (χ^2^/df < 5.0), Comparative Fit Index (CFI ≥ 0.95), Normed Fit Index (NFI ≥ 0.95), Incremental Fit Index (IFI ≥ 0.95), Tucker-Lewis Index (TLI ≥ 0.95), and Root-mean Square Error of Approximation (RMSEA < 0.08) [35]. The standardized estimates beta coefficient (β) and R^2^ were reported for the predictive strength and variances explained in the models, respectively. 

## 3. Results

All variables’ skewness and kurtosis values ranged from −1.04 to 0.65, which indicates a normal distribution among study variables. It was shown (Table 1) that participants had an average score of leisure-time PA (LTPA) that equals 58.73 (SD = 38.03) and a majority of these participants were physically active (72.2% of them had an LTPA score ≥ 24) [36]. Participants reported a moderate level of academic self-efficacy (M = 4.85, SD = 1.53 in a 7-Likert Scale), with 42.1% of adolescents reporting a score under 5 (from very disagree to fair). On average (Table 1), participants reported 20.72 (SD = 10.76) out of 60 on the depression scale, with 52.5% of them suspected to have depression (score ≥ 20) [37].

The correlation results were presented in Table 2. The significant and moderate associations among all intensity levels of PA were found in this study (*rs* range from 0.33 to 0.53, *p* < 0.01). The intensity levels of PA (LPA, MPA, VPA) had low but statistically significant positive associations with academic self-efficacy (*rs* range from 0.23 to 0.27, *p* < 0.01). Both MPA and VPA but not LPA were significantly and negatively correlated with depression (*p* < 0.01). Academic self-efficacy was significantly and negatively associated with depression (*r* = −0.39, *p* < 0.01).

The CFA analysis was conducted to test the measurement validity and reliability of the academic self-efficacy scale. The model fit indices suggested that the initial CFA model did not fit the data well (χ2/df = 92.01/5, *p* < 0.01, NFI = 0.96, IFI = 0.96, TLI = 0.92, CFI = 0.96, RMSEA = 0.20, 90% CI (0.17, 0.24)). Using the modification indices, three covariance between measurement errors of the academic self-efficacy items were added (items 1 and 2, items 3 and 4, and items 4 and 5). The goodness-of-fit indices (χ^2^/df = 7.54/2 = 3.77, *p* = 0.02, NFI = 1.00, IFI = 1.00, TLI = 0.99, CFI = 1.00, RMSEA = 0.08, 90% CI (0.03, 0.15)) suggested the model fit the data well after the error covariance were specified in the CFA model. Thus, the measurement model was preserved for the following structural model testing.

The SEM was structured to test the direct and indirect effects of different doses of PA (LPA, MPA, and VPA, exogenous variables) on depression (endogenous variable) through academic self-efficacy (latent/exogenous variable). The results (see Figure 1) produced sound goodness-of-fit indices (χ^2^/df = 47.63/18 = 2.64, *p* < 0.01, NFI = 0.98, IFI = 0.99, TLI = 0.98, CFI = 0.99, RMSEA = 0.06, 90% CI (0.04, 0.08)). It was found that LPA (β = 0.15, *p* < 0.01), MPA (β = 0.13, *p* < 0.05), and VPA (β = 0.12, *p* < 0.05) significantly predicted the academic self-efficacy and contributed 10% of variance in the model (*p* < 0.05). In the whole model, no significant direct effects were found from three doses of PA toward depression (*p* > 0.05), which suggests that 15% of the variance was contributed significantly from the path of academic self-efficacy (β = −0.37, *p* < 0.01). Thus, the results support that academic self-efficacy served as a full mediator in the three pathways from different doses of PA to depression among adolescents in this study.

## 4. Discussion

The main purpose of this study was to examine the relations between different doses of PA (LPA, MPA, and VPA), academic self-efficacy, and depression among middle school adolescents based on the psychosocial mechanism proposed by Lubans and colleagues [13]. The results supported the psychosocial mechanism that the different doses of PA (i.e., LPA, MPA, and VPA) had significant indirect effects on depression through academic self-efficacy among adolescents in China. These findings highlight the importance of PA on individuals’ confidence of academic achievements, which would ultimately justify their mental health outcomes, such as depression.

It is well known that, in Chinese culture, parents value academic achievement as a direct indicator of future success for their children [38]. In the current study, we found that more than half of our participants demonstrated depressive symptoms (depression scores ≥ 20), which suggests that the academic pressures are present at an age as young as 13 years old. Consistent with the hypothesis of Lubans and colleagues’ [13] conceptual model, the current study specifically supported the direct influence of academic self-efficacy on depression among adolescents [13]. However, the high rate of depression (52.5%) is alarming with higher than previous literature reported in Chinese adolescent populations [39]. Adolescence years are a crucial and critical time for developing the knowledge and skills to link childhood and adulthood characterized through significant psychological, emotional, and physical shifts. Levels of self-efficacy inversely influencing depression could have extreme implications to the mental health of these adolescents and have impacts on their physical well-being. 

It was well-documented that strong belief in one’s ability to complete certain tasks has a relatively consistent relationship with PA participation, especially engaging in MVPA among adolescents [18,26,40,41]. The current study confirmed that the significant and positive associations between different doses of PA (LPA, MPA, and VPA) and academic self-efficacy, and all intensity levels of PA were moderately correlated in this population. It is suggested that regularly engaging in different doses of PA such as chores around the house, brisk walking, biking, or running may positively influence adolescents’ academic self-efficacy. This finding also supports the notion that individuals who are physically active may engage in various types/levels of activities during the day, and participating in one level (e.g., LPA) does not compensate their participation in other levels. School health educators may provide various opportunities (i.e., recess, active classroom break, before and after school walking programs) for adolescents to cumulate recommended daily physical activity [15]. Building academic confidence and self-efficacy in adolescence is important. Strategies such as providing a supportive learning environment, positive feedback from parents, and less emphasis on a peer comparison on academic work are recommended in order to reduce the depressive symptoms.

The most unique contribution of this study was to provide the empirical evidence that any doses of PA (e.g., LPA) had direct or indirect benefits on depression prevention. Although only MVPA but not LPA were significantly correlated with depression in our sample, the SEM model supports the indirect effect from LPA to depression through academic self-efficacy. In other words, the indirect impact of LPA to depression through academic self-efficacy revealed that engaging in LPA is as beneficial as MPA and VPA to lower the risk of depression among adolescents. This is also in conjunction with recent evidence in other European countries (i.e., Spain), which showed that higher frequency of MVPA per day was associated with lower levels of depression and better well-being among adolescents [42,43]. The finding of our study provided preliminary evidence for the literature by testing the mediation effect of academic self-efficacy on the relationship between the different intensity levels of PA and depression among adolescents in China. That is, regardless of the intensity level, all forms/types of PA may contribute to adolescents’ mental health as well as their perceived academic self-efficacy. 

From a public health perspective, this study provided additional support that promoting the achievement of a daily 60-min PA guideline remains an effective strategy to improve adolescents’ health and wellness. Previous research found that adolescents tend to prefer a low-intensity level of PA, and individuals with depressive symptoms involve spending more time in inactivity [19,44]. With the results of this study, adolescents engaging in LPA (such as walking at a low intensity) could lower their depression levels by its direct effects on their academic self-efficacy. All intensity levels of PA are extremely important in mental health outcomes. However, light PA as a more attainable and feasible exercise intensity compared to MVPA should be promoted as an indirect way to prevent depression among adolescents. Specifically, health professions may design school-based physical activity programs/interventions by emphasizing a sense of accomplishment and pleasure with achievable and realistic goals to enhance individual’s self-efficacy and exercise compliance, which may reduce depression.

While the results of this study hold theoretical and practical appeal, there were several limitations. Due to the cross-sectional nature, the findings of this study preclude causal inferences of PA on depression through academic self-efficacy among adolescents in China. Longitudinal or experimental studies are needed to examine the possible causal nature of the cross-sectional associations that we have identified in this study. In addition, the current study relied exclusively on self-report measures, which are susceptible to response bias and distortion. Although the self-reported PA measure captures various forms of daily physical activities with specific metabolic equivalent of task (MET) scores, the objective measure of different doses of PA using accelerometers or heart rate monitors is also recommended in the future study. Lastly, the generalizability of these findings is limited given the fact that adolescents participating in this study were recruited from China. Future studies are needed to examine the relations between different doses of PA, academic self-efficacy, and depression among adolescents in other countries.

## 5. Conclusions

This study adds to a growing body of empirical evidence supporting the psychosocial mechanism of mental health proposed by Lubans and colleagues’ conceptual model [13]. Different doses of PA (i.e., LPA, MPA, VPA) were positively correlated with adolescents’ academic self-efficacy, therefore, reduce their risks of depression. The findings of this study suggest tailoring different intensities of PA benefits adolescents’ academic self-efficacy by framing the positive and supportive environment in schools (e.g., quality physical education, active recess), which can potentially reduce the prevalence of depression during adolescence.

## Figures and Tables

**Figure 1 children-07-00037-f001:**
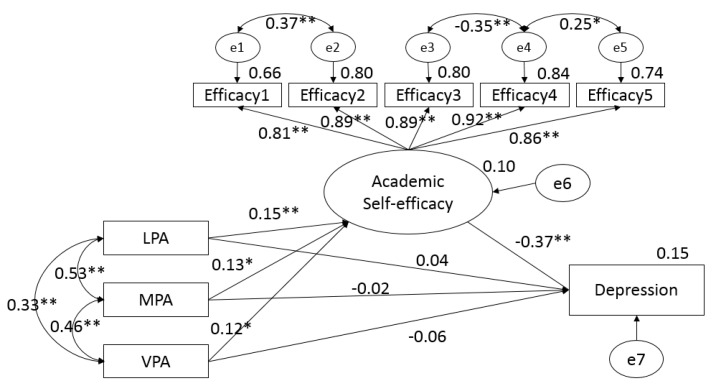
The results of the final atructural model. LPA = light physical activity, MPA = moderate physical activity, and VPA = vigorous physical activity. * *p* < 0.05, ** *p* < 0.01.

**Table 1 children-07-00037-t001:** The descriptive statistics of study variables (*n* = 428).

Variables	Score Range	Total Sample (*n* = 428) M (SD)	Girls (*n* = 193) M (SD)	Boys (*n* = 235) M (SD)
Academic Self-efficacy	1–7	4.85 (1.53)	4.67 (1.29)	5.00 (1.70)
Depression	1–60	20.72 (10.76)	21.83 (11.08)	19.75 (10.77)
LPA (MET/week)	0–24	13.41 (7.43)	13.25 (6.87)	13.54 (7.88)
MPA (MET/week)	0–40	19.50 (12.61)	17.95 (11.93)	20.78 (13.02)
VPA (MET/week)	0–72	26.66 (26.01)	26.34 (25.78)	26.92 (26.26)
LTPA (MET/week)	0–136	58.73 (38.03)	56.64 (36.05)	60.46 (39.58)
LTPA ≥ 24 (MET/week)	-	309(72.2%)	133 (69%)	176 (74.9%)

LPA = light physical activity. MPA = moderate physical activity. VPA = vigorous physical activity. LTPA = leisure time physical activity.

**Table 2 children-07-00037-t002:** Correlations among study variables.

Variables	1	2	3	4	5
1. Academic Self-efficacy	-				
2. Depression	−0.39 *	-			
3. LPA	0.25 *	−0.09	-		
4. MPA	0.27 *	−0.13 *	0.53 *	-	
5. VPA	0.23 *	−0.15 *	0.33 *	0.46 *	-

** p* < 0.01.

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
