# Peer review of "Psychosocial Mechanism of Adolescents’ Depression: A Dose-Response Relation with Physical Activity"

_children, 2020, doi:10.3390/children7040037_

Round 1

Reviewer 1 Report

This manuscript addresses a cross-sectional study regarding the association between different doses of physical activity and depressive symptoms, mediated by self-efficacy at school. Generally, I think it`s a well-formulated and novel study; however, I have several minor concerns listed as following:

  1. In the first paragraph of “Introduction”, the authors reviewed abundant information about depression; however, I think the terms” depression” is not precise. Please clarify if it is depressive disorder or depressive symptoms?
  2. Please give the IRB approved numbers in the section “Materials and Methods”.
  3. In the section 2.3.2, please give the details of efficacy 1 to 5.
  4. In the Table.1, add additional information regrading the distribution of population from LPA to VPA.
  5. In the figure.1, the authors estimated the correlation between LPA, MPA, and VPA. Is there any clinical relevance for this correlation? Again, please explain the correlation marked on the residuals.
  6. In the 4th paragraph of “Discussion”, I am interested in which dose of physical activity is most beneficial to prevent depressive symptoms for adolescents.

Author Response

Responses to Reviewers’ Comments

Manuscript ID: children-767284. Psychosocial Mechanism of Adolescents’ Depression: A Dose-Response Relation with Physical Activity

Reviewer -1 Comments

This manuscript addresses a cross-sectional study regarding the association between different doses of physical activity and depressive symptoms, mediated by self-efficacy at school. Generally, I think it`s a well-formulated and novel study; however, I have several minor concerns listed as following:

Response: Thank you for the favorable feedback. In this round of revision, we have carefully reviewed and addressed all comments.

In the first paragraph of “Introduction”, the authors reviewed abundant information about depression; however, I think the terms” depression” is not precise. Please clarify if it is depressive disorder or depressive symptoms?

Response: We have clarified the depression in a way of depressive symptoms in the first paragraph. Line 33-40.

Please give the IRB approved numbers in the section “Materials and Methods”.

Response: The information has been added. Line 102.

In the section 2.3.2, please give the details of efficacy 1 to 5.

Response: The detailed information has been added. Line 132-134.

In the Table.1, add additional information regarding the distribution of population from LPA to VPA.

Response: The Table -1 has been updated accordingly. According to the LTPA protocol, there were 72.2% of the participants are active with MET score ≥ 24, and around 69% active girls and 74.9% active boys in this sample.

In the figure.1, the authors estimated the correlation between LPA, MPA, and VPA. Is there any clinical relevance for this correlation? Again, please explain the correlation marked on the residuals.

Response: The correlations among LPA, MPA, and VPA have been presented in the results section and the clinical implication have been added in the discussion accordingly. Line 171-172 and line 224-228.

Since the CFA was not the interest of this study, we did not explain the correlation/covariance of the residuals in the paper. It represents covariance (or correlation) between the factors that is not explained by the predictors. It means that two factors are causally related. In order to address your comment here, detailed procedures regarding the CFA were added with specific information of three covariance between measurements errors. Line 181-188.

In the 4th paragraph of “Discussion”, I am interested in which dose of physical activity is most beneficial to prevent depressive symptoms for adolescents.

Response: The paragraph has been revised and the specific insights/suggestion have also been discussed in the following paragraph. Line 237-248 and line 251-260.

Reviewer 2 Report

Peer review of manuscript ID children-767284
Title: Psychosocial Mechanism of Adolescents’ Depression: A Dose-Response Relation with Physical Activity

The study is interesting and well thought out, although citations in text are recommended for a better understanding, even if they are carried out according to the journal's regulations, including the author and the year and, next to it, the call that leads to the references. It would also be convenient, for subsequent studies, to review references from the Hispanic field, because,
specifically in Spain, quite a few studies have been carried out in this regard.

Author Response

Reviewer -2 Comments

Title: Psychosocial Mechanism of Adolescents’ Depression: A Dose-Response Relation with Physical Activity

The study is interesting and well thought out, although citations in text are recommended for a better understanding, even if they are carried out according to the journal's regulations, including the author and the year and, next to it, the call that leads to the references. It would also be convenient, for subsequent studies, to review references from the Hispanic field, because, specifically in Spain, quite a few studies have been carried out in this regard.

Response: The reference and the format of the paper have been corrected.

The relevant studies have been included in the discussion accordingly. Line 241-244.

Thank you for the favorable feedback. In this round of revision, we have carefully reviewed and addressed all comments.
